# Glucose uptake to guard cells via STP transporters provides carbon sources for stomatal opening and plant growth

Sabrina Flütsch[1,2,†] (ID), Arianna Nigro[2,†,‡], Franco Conci[2], Jiří Fajkus[3], Matthias Thalmann[2,§] (ID), Martin Trtílek[3], Klára Panzarová[3] & Diana Santelia[1,2,*] (ID)

## Abstract

Guard cells on the leaf epidermis regulate stomatal opening for gas exchange between plants and the atmosphere, allowing a balance between photosynthesis and transpiration. Given that guard cells possess several characteristics of sink tissues, their metabolic activities should largely depend on mesophyll-derived sugars. Early biochemical studies revealed sugar uptake into guard cells. However, the transporters that are involved and their relative contribution to guard cell function are not yet known. Here, we identified the monosaccharide/proton symporters Sugar Transport Protein 1 and 4 (STP1 and STP4) as the major plasma membrane hexose sugar transporters in the guard cells of *Arabidopsis thaliana*. We show that their combined action is required for glucose import to guard cells, providing carbon sources for starch accumulation and light-induced stomatal opening that are essential for plant growth. These findings highlight mesophyll-derived glucose as an important metabolite connecting stomatal movements with photosynthesis.

**Keywords** glucose; guard cells; plant growth; stomatal opening; sugar transport protein
**Subject Categories** Metabolism; Plant Biology

## Introduction

Stomata are microscopic pores on the plant leaf epidermis surrounded by a pair of guard cells. These vital cells adjust pore aperture in response to numerous endogenous and exogenous factors, allowing uptake of carbon dioxide ($CO_2$) for photosynthesis ($A$), while preventing excessive water loss through transpiration ($E$). By controlling the trade-off between photosynthesis and transpiration, stomata play a critical role in determining water-use efficiency (WUE = amount of carbon fixed per unit water loss, $A/E$) and, hence, plant growth and productivity (Lawson & Vialet-Chabrand, 2019).

Stomatal opening and closure results from reversible changes in guard cell volume and shape. At the molecular level, this is driven primarily by the activity of the plasma membrane $H^+$-ATPase which stimulates the movement of large quantities of ions (mainly potassium, chloride, malate$^{2-}$, and nitrate) into and out of the guard cells and consequent osmotic water flow for their swelling or shrinking (Inoue & Kinoshita, 2017; Jezek & Blatt, 2017).

For more than a century, it has been known that guard cells possess modified carbohydrate metabolic pathways compared to the rest of the leaf, but their significance for stomatal function has long remained obscure. Only recently it became clear that guard cell starch metabolism integrates with signaling and membrane ion transport to regulate stomatal movements (Daloso *et al*, 2017; Santelia & Lawson, 2016). At the start of the day, the rapid breakdown of guard cell starch is activated by phototropin-mediated signaling downstream of the plasma membrane $H^+$-ATPase to promote efficient stomatal opening (Horrer *et al*, 2016). The major guard cell starch-derived metabolite is glucose (Glc), which is needed to maintain the cytoplasmic sugar pool contributing to fast stomatal opening (Flütsch *et al*, 2020). Starch formation induced during high $CO_2$-mediated stomatal closure has been proposed to facilitate the dissipation of the accumulated organic solutes leading to changes in guard cell osmotic potential for water efflux (Penfield *et al*, 2012; Azoulay-Shemer *et al*, 2016). Soluble sugars also regulate stomatal movements, but the way they do so remains still rather controversial (Daloso *et al*, 2016a). Initial studies suggested that sucrose (Suc) and its derived sugars (Glc; fructose, Fru) induce stomatal opening as direct osmotica (Outlaw & Manchester, 1979; Poffenroth *et al*, 1992; Talbott & Zeiger, 1993; Amodeo *et al*, 1996). More recently, it was revealed that Suc promotes stomatal opening

1  Institute of Integrative Biology, ETH Zürich, Zürich, Switzerland
2  Department of Plant and Microbial Biology, University of Zürich, Zürich, Switzerland
3  Photon Systems Instruments (PSI), Drasov, Czech Republic
   *Corresponding author. Tel: +41 44 632 89 27; E-mail: dsantelia@ethz.ch
   †These authors contributed equally to this work
   ‡Present address: Syngenta Crop Protection AG, Stein AG, Switzerland
   §Present address: John Innes Centre, Norwich Research Park, Norwich, UK

by serving as a substrate for glycolysis and mitochondrial respiration (Daloso *et al*, 2015, 2016b; Medeiros *et al*, 2018). Sugars can also induce stomatal closure either as osmolytes in the guard cell apoplast (Lu *et al*, 1997; Outlaw & De Vlieghere-He, 2001; Kang *et al*, 2007) or as signaling molecules through phosphorylation by hexokinase within the guard cells (Kelly *et al*, 2013; Lugassi *et al*, 2015).

Despite these studies shed light on the importance of carbohydrate metabolism for stomatal movements, experiments have not yet answered basic questions about the source of sugars in guard cells. Given that $CO_2$ fixation within guard cells is limited (Outlaw *et al*, 1979; Outlaw, 1989; Reckmann *et al*, 1990) and that photosynthesis in the mesophyll cells is the main source of sugars at the whole-plant level, it is likely that symplastically isolated guard cells rely mostly on mesophyll-derived Suc to fulfill their metabolic needs. Suc can be taken up directly via Suc transporters or in the form of hexoses via monosaccharide transporters following Suc hydrolysis by a cell wall invertase. Transcriptomics studies suggest that several sugar transporters are highly expressed in guard cells (Leonhardt *et al*, 2004; Wang *et al*, 2011; Bates *et al*, 2012; Bauer *et al*, 2013), but the relative contribution of this transport system to guard cell function is not yet known. Here, we identified the monosaccharide/proton symporters Sugar Transport Protein 1 and 4 (STP1 and STP4) as the major plasma membrane hexose sugar transporters in the guard cells of *Arabidopsis thaliana*. We show that their combined action is required for Glc import to guard cells, providing carbon sources for starch accumulation and light-induced stomatal opening that are essential for plant growth. These findings highlight that a tight coordination between mesophyll and guard cell carbohydrate metabolism promotes optimal plant growth through regulation of stomatal opening.

# Results

## Sugar Transport Protein 1, 4, and 13 are highly expressed in guard cells

Higher plants possess three types of plasma membrane carriers for the intercellular transport of sugars: MSTs (monosaccharide transporters), SUCs or SUTs (Suc transporters) and SWEETs (hexose and Suc transporters). Angiosperm genomes usually contain several paralogs of each class of transporters, most of which serve distinct physiological roles (Chen *et al*, 2015).

Through literature and database searches, we identified 40 plasma membrane sugar transporters in the *Arabidopsis* genome, covering all three types of carriers (Appendix Table S1). To select potential candidates for our study, we performed *in silico* analysis of gene expression levels in *Arabidopsis* guard cells using publicly available expression data (Fig EV1A). As expected, several transporters were highly expressed in guard cells, for instance, sucrose transporters 1, 2, and 3 (*SUC1*, *SUC2*, *SUC3*); Sugars will eventually be exported transporters 1, 5, 11, and 12 (*SWEET1*, *SWEET5*, *SWEET11*, *SWEET12*); sugar transport proteins 1, 4, 5, and 13 (*STP1*, *STP4*, *STP5*, *STP13*); and polyol/monosaccharide transporters 4, 5 and 6 (*PMT4*, *PMT5*, *PMT6*) (Fig EV1A). We focused on STP1, 4, and 13, as their gene expression in guard cells was on

average 15 to 40 times higher compared to other sugar transporters (Fig EV1A). We confirmed their high and preferential expression in guard cell-enriched epidermal peels by qPCR (Fig EV1B and Appendix Table S2). In a previous study, STP1 was shown by *in situ* hybridization and immunohistochemistry to localize to guard cells (Stadler *et al*, 2003), further supporting our results.

STPs are high-affinity monosaccharide/proton symporters responsible for the transport of Glc, Fru, galactose, mannose, arabinose, and xylose from the apoplastic space into the cytosol (Büttner & Sauer, 2000; Büttner, 2010; Poschet *et al*, 2010; Rottmann *et al*, 2016, 2018b). These transporters are mostly found in sink tissues or symplastically isolated cells, such as pollen tubes, developing embryo, or guard cells (Stadler *et al*, 2003; Büttner, 2010; Rottmann *et al*, 2018a). It has been shown that STPs fulfill three main functions in plants: uptake of monosaccharides for the nutrition of sink cells (Sherson *et al*, 2000); re-absorption of monosaccharides from damaged roots under abiotic stress (Yamada *et al*, 2011); and antibacterial defense by competing with pathogens for extracellular sugars (Yamada *et al*, 2016).

## Light-induced stomatal opening is impaired in *Arabidopsis* plants lacking both STP1 and STP4 transporters

To assess the contribution of the selected STPs to stomatal function, we obtained homozygous *Arabidopsis* T-DNA insertion lines at the *STP1* (*stp1-1*; SALK_048848 and *stp1-2;* SALK_139194), *STP4* (*stp4-1*; SALK_049432 and *stp4-2;* SALK_091229) and *STP13* (*stp13*, SALK_0455494) loci. qPCR analyses revealed disruption of the *STP1* gene expression in the *stp1-1* mutant line (Fig EV1C), and a reduction of *STP1* transcripts of 60% in the *stp1-2* mutant (Fig EV1D). Furthermore, *STP4* and *STP13* transcript levels were reduced by approximately 40 and 80% in their respective mutant backgrounds compared to wild type (WT; Fig EV1C and D). To uncover putative functional relationship between the different STP isoforms, we generated the double mutant combinations *stp1stp4* (from *stp1-1* and *stp4-1*), *stp1stp13* (from *stp1-1*), and *stp4stp13* (from *stp4-1*).

To describe morpho-physiological performance of the mutant lines *in vivo*, we used the automated phenotyping platform Plant-Screen™ Compact System (PSI, Czech Republic). We established a robust phenotyping protocol to quantify daily, over a period of 8 days, plant morphological, physiological, and biochemical traits. Infrared thermography revealed that *stp1stp4* plants had statistically significant higher leaf surface temperature compared to WT and all tested mutant combinations, even though the overall differences in surface temperatures were small (Fig 1A and B; Appendix Table S3). Given that leaf temperature is an indicator of stomatal aperture (Merlot *et al*, 2002), we hypothesized that *stp1stp4* mutant plants may have closed stomata. Indeed, infrared gas analysis of stomatal conductance ($g_s$) responses showed that light-induced stomatal opening was severely impaired in *stp1stp4* plants (Fig 1C). Stomatal closure in response to darkness was also affected in this mutant (Fig 1C). The *stp1-1* single mutant had a reduced steady-state $g_s$. However, *stp1-1* plants reached a similar overall $g_s$ amplitude as WT, but stomatal opening kinetics were slow (Fig 1C), well visible if $g_s$ values were normalized to values at the end of the night (EoN; Fig EV2A). The slow opening phenotype of *stp1-1* single mutants was further confirmed in a second mutant allele *stp1-2*

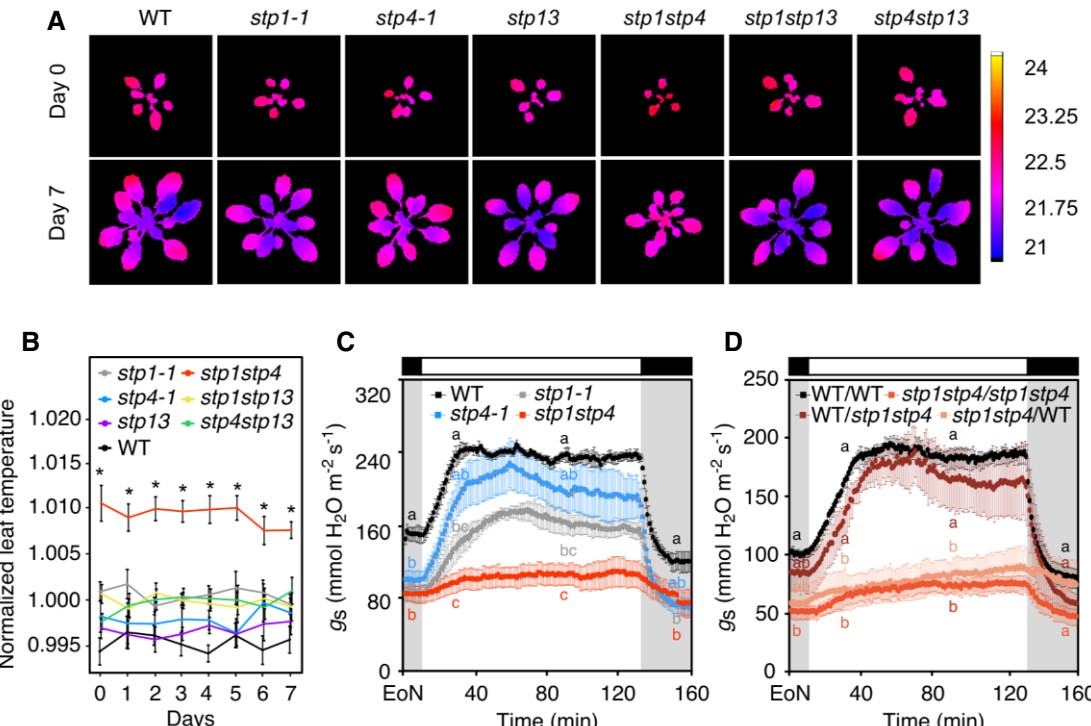

**Figure 1. Stomatal function is impaired in *stp1stp4* plants.**

A   Representative false color images of leaf surface temperature captured by a thermal camera from WT, *stp1-1*, *stp4-1*, *stp1stp4*, *stp13*, *stp1stp13*, and *stp4stp13* plants.

B   Normalized leaf surface temperature over the phenotyping period. Data shown are means ± SEM; $n = 10$ per genotype and time point.

C   Whole-plant recordings of changes in stomatal conductance ($g_s$) from WT, *stp1-1*, *stp4-1*, and *stp1stp4* plants. Data shown are means ± SEM; $n \geq 3$ per genotype.

D   Whole-plant recordings of changes in stomatal conductance ($g_s$) from self-grafted donor lines (WT/WT, *stp1stp4*/*stp1stp4*) and reciprocal grafting of shoot/root (WT/*stp1stp4*, *stp1stp4*/WT) plants. Data shown are means ± SEM; $n = 3$ per genotype.

Data information: (C, D) Plants have been illuminated with 150 µmol/m²/s white light after the end of the night (EoN) under ambient-air $CO_2$ concentrations. (B) Asterisk (*) indicates significant statistical difference between WT and *stp* plants for $P < 0.05$ determined by one-way ANOVA with post hoc Tukey's test. (C, D) Different letters indicate significant statistical differences among genotypes for the given time point for $P < 0.05$ determined by one-way ANOVA with post hoc Tukey's test.

(Fig EV2C and D). The mild stomatal opening phenotype of *stp1* mutants can be explained by a strong upregulation of *STP13* in the guard cells of mutant plants (Appendix Fig S1). STP13 might partially compensate for the loss of STP1 in the *stp1* mutant. Interestingly, *stp4-1* single mutants also had a reduced steady-state $g_s$, but reached a greater overall $g_s$ amplitude compared to WT plants and showed similar stomatal opening kinetics (Figs 1C and EV2A). In addition, *stp4-2* showed a similar elevated $g_s$ amplitude as the *stp4-1* (Fig EV2C and D), indicating that mutation in the *stp4* locus is responsible for the observed phenotype. Altogether, the phenotype of the single *stp1-1* and *stp4-1* mutants and their respective additional mutant alleles (*stp1-2* and *stp4-2*), with $g_s$ amplitudes and stomatal opening kinetics similar to WT, suggests that STP1 and STP4 are both required to promote stomatal opening at the start of the day (Figs 1A–C and EV2A, C and D; Appendix Table S3). Despite the high expression of *STP13* in guard cells (Fig EV1), the lack of functional STP13 in the *stp13* single mutant did not cause a reduced $g_s$ amplitude nor slow opening kinetics. *Stp13* mutants behaved similar to the *stp4* mutant alleles (Figs 1A and B, and EV2E and F; Appendix Table S3). To investigate possible reasons behind the lack of phenotypes in *stp4*, *stp13*, *stp1stp13*, and *stp4stp13*, we performed *STP* gene expression analyses on guard cell-enriched

epidermal peels of WT, *stp4-1*, and *stp13* plants. Intriguingly, we found that *STP1* was upregulated in guard cells of *stp4-1* and *stp13* plants (Appendix Fig S1), suggesting that STP1 might partially take over the role of STP4 and STP13 in their absence. In addition, *STP13* is upregulated in response to pathogen infections or treatments with bacterial elicitors (Büttner, 2010), and it is the only *STP* gene inducible by osmotic stress, high salinity and abscisic acid (Yamada *et al*, 2011). STP13 function in guard cells may therefore become critical under stress conditions.

Previous studies reported the expression of *STP1* and *STP4* in *Arabidopsis* roots and their involvement in the uptake of monosaccharides from the rhizosphere (Truernit *et al*, 1996; Sherson *et al*, 2000; Yamada *et al*, 2011). To rule out the possibility that simultaneous knock-out of *STP1* and *STP4* in the roots contributed to the severe impairment of stomatal opening in *stp1stp4*, we measured $g_s$ in WT/*stp1stp4* and *stp1stp4*/WT grafted plants. Plants with WT shoots and *stp1stp4* roots showed stomatal conductance comparable to WT, whereas plants bearing *stp1stp4* shoots and WT roots displayed a *stp1stp4*-like phenotype (Figs 1D and EV2B). The genetic identity of the roots from the reciprocal grafted plants was confirmed by molecular genotyping (Fig EV2G). Our grafting experiments indicate that the stomatal phenotype of *stp1stp4* is

independent from the function of these transporters in the roots and further support their essential role in guard cells.

### stp1stp4 guard cells have reduced levels of glucose

We reasoned that inhibition of light-induced stomatal opening in *stp1stp4* mutants might be a direct consequence of impaired Glc and/or Fru import to guard cells. To test this hypothesis, we measured soluble sugar content in guard cells of intact leaves of WT and *stp1stp4* mutant plants at the EoN and after 40 min of light (Fig 2A). In WT guard cells, the levels of Glc and Suc were unaltered in response to the light treatment, while Fru levels decreased to half due to illumination (Fig 2A). The sustained levels of Suc are likely due to continuous Suc import from the mesophyll, which is consistent with the high expression of *SUC1* and *SUC3* transporters in guard cells at the EoN (Fig EV3A). Notably, *stp1stp4* guard cells had significantly lower amounts of Glc and Fru at the EoN compared to WT, and Glc levels were almost undetectable after 40 min of light (Fig 2A). Suc surprisingly accumulated to higher levels (Fig 2A), perhaps as a result of reduced consumption, sequestration in the vacuole or *SUC* transporters upregulation to compensate for the loss of STPs. In contrast to *stp1stp4* double mutants, guard cells of single *stp1-1* and *stp4-1* mutants contained similar amounts of sugars as that of WT guard cells (Fig EV3B). However, there was a trend in *stp1-1* guard cells towards reduced Glc levels compared to WT (Fig EV3B), which is in line with the mild impairment in stomatal opening in these plants.

These data demonstrate *in vivo* the function of STP1 and STP4 in the coordinate transport of Glc and to a lesser extent Fru at the guard cell plasma membrane during light-induced stomatal opening and likely explain the inability of *stp1stp4* to open stomata in the light. Based on the phenotype of the *stp1stp4* mutant, which cannot open stomata despite the high levels of Suc, we suggest that imported Glc provides a major source of carbon for light-induced stomatal opening at the start of the day.

### stp1stp4 guard cells are devoid of starch

Given that guard cells possess several characteristics of sink tissues with fewer and smaller chloroplasts and low photosynthetic rates, we next investigated whether the import of apoplastic Glc by STPs influenced guard cell starch metabolism. As we reported previously (Horrer *et al*, 2016), starch was degraded in WT guard cells when the plants were illuminated, coinciding with the opening of the stomata (Fig 2B and C); after falling to near zero in the first hour after dawn, starch levels then began to rise again (Fig 2B and C). Notably, at the EoN, *stp1stp4* guard cells were essentially devoid of starch, and no starch synthesis occurred during the first 3 h of light (Fig 2B and C). The single *stp* mutants displayed a milder phenotype. Although they degraded the starch in guard cells upon transition to light similar to WT, they failed to resynthesize it (Figs 2B and C, and EV3C and D).

The fact that *stp1stp4* guard cells were unable to make starch even in the presence of high amounts of Suc led us to hypothesize that at the start of the day mesophyll-derived Glc imported by STPs

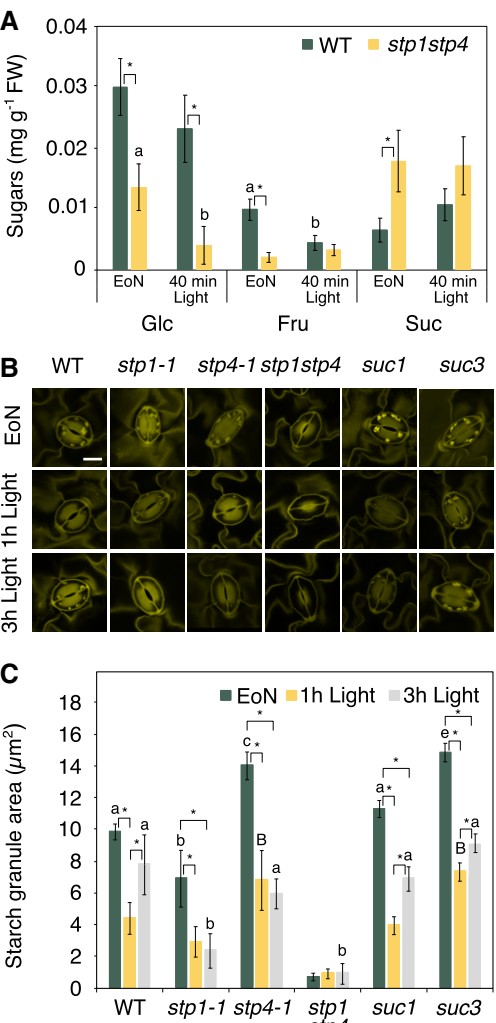

**Figure 2. *stp1stp4* plants have low levels of hexoses and are devoid of starch in guard cells.**

A  Content of soluble sugars in guard cell-enriched epidermal peels of WT and *stp1stp4* plants at the end of the night (EoN) and after 40 min of illumination with white light at 150 μmol/m²/s following the EoN. Data for two independent experiments are shown; means ± SEM; *n* ≥ 7 per genotype and time point.

B  Representative confocal laser microscopy images of propidium iodide-stained guard cell starch granules of intact leaves of WT, *stp1-1, stp4-1, stp1stp4, suc1, and suc3* plants. Scale bar, 10 μm.

C  Starch dynamics in guard cells of intact leaves of WT, *stp1-1, stp4-1, stp1stp4, suc1, and suc3* plants at the end of the night (EoN) and after 1 and 3 h of illumination with 150 μmol/m²/s of white light. Data for three independent experiments are shown; means ± SEM; *n* = 120 individual guard cells per genotype and time point.

Data information: (A) Different letters indicate significant statistical differences among time points for the given genotype. Asterisk (*) indicates significant statistical difference between genotypes for the given time point for *P*< 0.05 determined by one-way ANOVA with post hoc Tukey's test. (C) Different letters indicate significant statistical differences among genotypes for the given time point. Asterisk (*) indicates significant statistical difference among time points for the given genotypes for *P* < 0.05 determined by one-way ANOVA with post hoc Tukey's test.

is the precursor for guard cell starch biosynthesis. If this hypothesis is correct, *Arabidopsis* plants lacking SUC transporters should be able to make starch under the investigated conditions. Indeed, as we show in Fig 2B and C, guard cells of *suc1* and *suc3* mutants had essentially normal starch turnover during the first 3 h of light (Fig 2B and C).

### *stp1stp4* stomatal phenotype impacts on plant photosynthesis and growth

Nearly all $CO_2$ needed for mesophyll photosynthesis enters the plant waxy leaf epidermis through the stomatal pores. Consequently, we predicted that reduced stomatal aperture in *stp1stp4* should have a major impact on intercellular $CO_2$ concentrations ($C_i$, μmol $CO_2$ mol/air) and photosynthetic performance. As anticipated, estimations of $C_i$ of illuminated leaves were lower in *stp1stp4* compared to WT and the single *stp* mutants (Fig EV4A). Kinetic chlorophyll fluorescence imaging further revealed that photosystem II (PSII) operating efficiency ($\Phi_{PSII}$) (Figs 3A and B, and EV4B and C), maximum quantum yield of PSII photochemistry in the dark-adapted ($F_v/F_m$) and the light-adapted ($F_v'/F_m'$) states, and photochemical quenching (qP) (Appendix Table S4) were reduced in *stp1stp4* compared to WT and all tested mutant combinations. Independent measurements of $CO_2$ assimilation rates ($A$) confirmed these observations, both for mutant and grafted plants (Figs 3C and D, and EV4D–I). Consistent with the reduced rate of photosynthesis, we found that Glc, Fru, and Suc levels in *stp1stp4* leaves at the end of the day (EoD) were, respectively, 40, 60, and 65% lower than that of WT (Fig 3E–G). EoD leaf starch content was also reduced by 60%, indicating that *stp1stp4* failed to accumulate starch efficiently in mesophyll cell chloroplasts (Fig 3H). In contrast to *stp1stp4*, the single *stp* mutants and the other *stp* double mutant combinations had sugar and starch levels similar to that of WT (Figs 3E–G and EV4J–M).

Using top-view RGB imaging, we determined growth performance of the plants. The *stp1stp4* mutants had 60% reduction in growth and a lower growth rate compared to WT (0.175 versus 0.211 per day; Appendix Table S5), while *stp1-1*, *stp4-1*, and *stp13* showed an intermediate phenotype, with only 20% reduction in growth (Figs 3I and J, and EV4N and O; Appendix Table S5). The *stp1stp13* and *stp4stp13* double mutants were indistinguishable from WT (Fig EV4N and O; Appendix Table S5); this again could be explained by a compensating mechanism between different STP isoforms when one is missing (Appendix Fig S1). No morphological differences among WT and *stp* mutants were observed, with the exception of reduced rosette perimeter and leaf slenderness in the case of *stp1stp4* (Appendix Table S5).

### Elevated ambient-air $CO_2$ restores *stp1stp4* leaf carbohydrate metabolism and photosynthetic growth

Diffusional $CO_2$ limitation on $A$ imposed by stomatal conductance can be balanced off by subjecting plants to elevated ambient-air $CO_2$. With this in mind, we reasoned that exposing *stp1stp4* to high external $CO_2$ concentrations would improve its photosynthetic capacity. Notably, when ambient-air $CO_2$ was increased from 400 to 600–1,000 ppm, *stp1stp4* steady-state $A$ values fully recovered and were even slightly elevated compared to that of WT (Fig 4A).

Furthermore, *stp1stp4* plants grown in controlled environment under 600 ppm $CO_2$ accumulated WT levels of leaf soluble sugars and starch (Fig 4B–E) and displayed a growth rate similar to that of WT (0.17 versus 0.18 per day; Fig 4F and G). The high $CO_2$ treatment also increased starch accumulation in guard cells of WT at the EoN, but had no effect on *stp1stp4* guard cells, which were still devoid of starch (Fig 4H and I). The lack of starch in *stp1stp4* guard cells under elevated $CO_2$ suggests that this phenotype is a direct consequence of the genetic defect in this mutant and further supports our conclusion that starch in guard cells is primarily made using mesophyll-derived Glc imported via STPs.

Overall, these results indicate that elevated $CO_2$ concentrations could restore *stp1stp4* leaf carbohydrate metabolism and photosynthetic growth to WT levels, implying the presence of a fully functional photosynthetic apparatus in *stp1stp4*. Indeed, measurements by visible-near-infrared (VNIR) hyperspectral imaging of Normalized Difference Vegetation Index (NDVI), a commonly used estimator of chlorophyll content (Rouse *et al*, 1974), indicated that NDVI of *stp1stp4* plants was comparable to that of WT and the other *stp* mutants (Fig EV5A and B; Appendix Table S3). Moreover, estimation of the variation in rosette green colors using the greenness hue abundance automatically computed from color-segmented RGB images showed no significant differences in paler hues of green (3, 4, and 5) for *stp* plants and between *stp1stp4* and WT plants (Fig EV5C–I; Appendix Table S5). Chlorophyll content was also unchanged (Fig EV5J).

Based on these observations, we conclude that simultaneous mutation of *STP1* and *STP4* genes in *Arabidopsis* results in diffusive stomatal limitation that constrains $CO_2$ availability for mesophyll photosynthesis, explaining the low photosynthetic performance, the alterations in mesophyll carbohydrate metabolism, and the resulting defective growth phenotype of *stp1stp4* mutant.

## Discussion

Due to limited autonomous $CO_2$ fixation capacity (Outlaw, 1989; Reckmann *et al*, 1990) and the lack of functional plasmodesmata (Wille & Lucas, 1984), it was long hypothesized that guard cells import mesophyll-derived sugars as a source of nutrition and osmotica. A few $^{14}$C pulse-labeling studies have documented sugar uptake into guard cells, providing some evidence that it occurs in symport with $H^+$ in a process that is energy dependent, enhanced in the presence of fusicoccin (an activator of the plasma membrane $H^+$-ATPase), and inhibited by uncouplers (Dittrich & Raschke, 1977; Reddy & Das, 1986; Lu *et al*, 1995, 1997; Ritte *et al*, 1999). In these early experiments, the identity of the involved sugar transporters had not been determined, and our current knowledge is still mostly based on transcriptomic studies. The "omics" approach has provided a long list of sucrose and hexose transporter genes that are highly expressed in guard cells compared to mesophyll cells (Leonhardt *et al*, 2004; Wang *et al*, 2011; Bates *et al*, 2012; Bauer *et al*, 2013), but experimental evidence to support their function in guard cell regulation is essentially lacking.

In the present study, we identified the high-affinity monosaccharide/$H^+$ symporters STP1 and STP4 as major guard cell plasma membrane hexose transporters in *Arabidopsis*. We provide genetic, physiological and biochemical evidence that STP1 and STP4

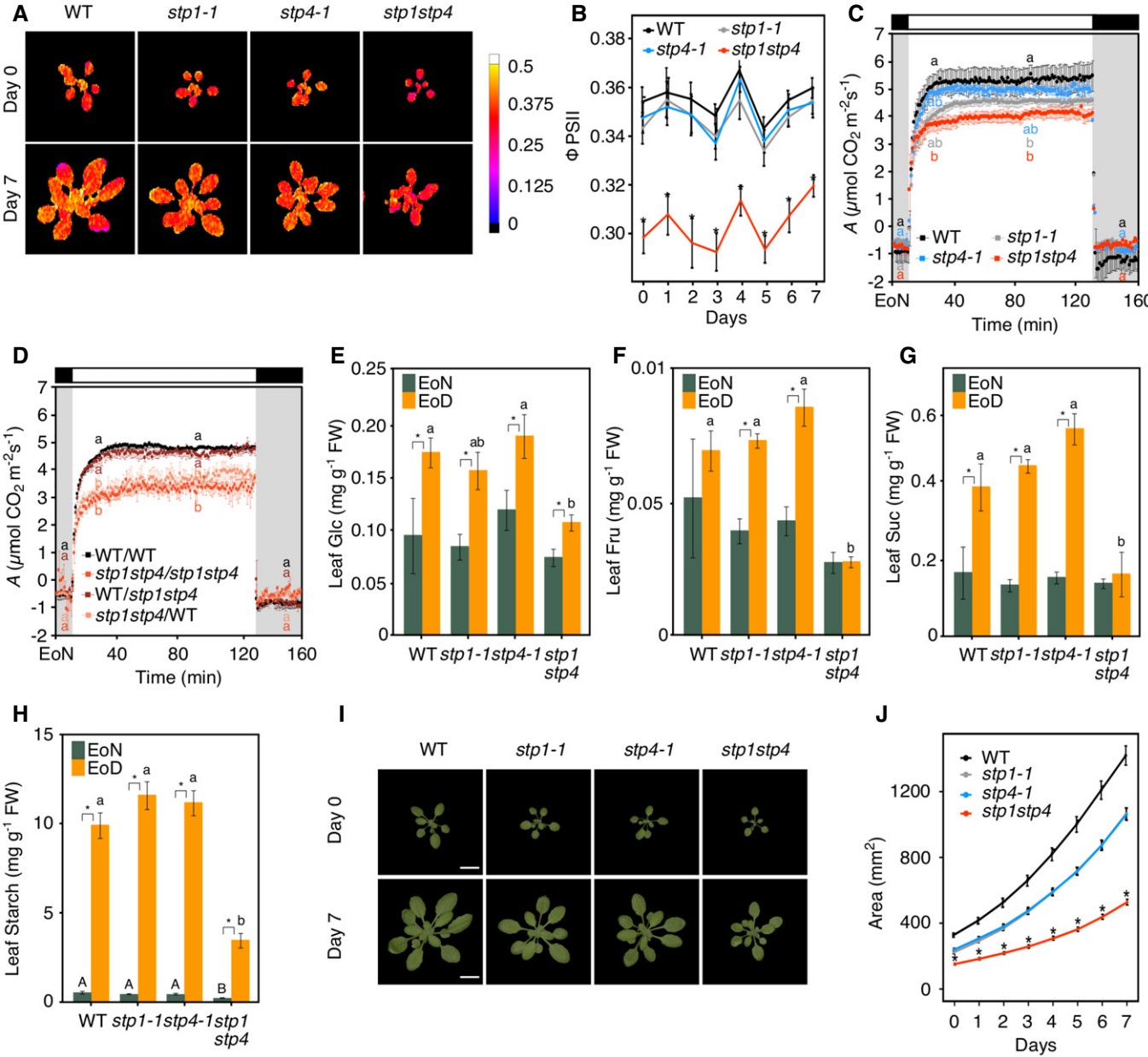

**Figure 3.  *stp1stp4* plants have reduced photosynthetic growth.**

A   Representative false color images of photosystem II (PSII) operating efficiency (ΦPSII) captured by chlorophyll fluorescence imaging from WT, *stp1-1*, *stp4-1* and
    *stp1stp4* plants. ΦPSII was measured at a photosynthetically active radiation (PAR) of 440 μmol/m$^2$/s.
B   ΦPSII quantified over the phenotyping period. Data shown are means ± SEM; $n$ = 10 per genotype and time point.
C   Whole-plant recordings of changes in $CO_2$ assimilation (*A*) from WT, *stp1-1*, *stp4-1*, and *stp1stp4* plants. Data shown are means ± SEM; $n \geq 3$ per genotype.
D   Whole-plant recordings of changes in $CO_2$ assimilation (*A*) from self-grafted donor lines (WT/WT, *stp1stp4*/*stp1stp4*) and reciprocal grafting of shoot/root (WT/
    *stp1stp4*, *stp1stp4*/WT) plants. Data shown are means ± SEM; $n$ = 3 per genotype.
E   Quantification of leaf glucose (Glc). Data shown are means ± SEM; $n$ = 8 per genotype and time point.
F   Quantification of leaf fructose (Fru). Data shown are means ± SEM; $n$ = 8 per genotype and time point.
G   Quantification of leaf sucrose (Suc). Data shown are means ± SEM; $n$ = 8 per genotype and time point.
H   Quantification of leaf starch. Data shown are means ± SEM; $n$ = 8 per genotype and time point.
I   Representative Red Green Blue (RGB) images of 3-(day 0) and 4-week-old (day 7) WT, *stp1-1*, *stp4-1*, and *stp1stp4* plants. Scale bar, 10 μm.
J   Projected rosette area over the phenotyping period. Data shown are means ± SEM; $n$ = 10 per genotype and time point.

Data information: (B and J) Asterisk (*) indicates significant statistical difference between WT and *stp* plants for $P < 0.05$ determined by one-way ANOVA with post hoc
Tukey's test. (C and D) Plants have been illuminated with 150 μmol/m$^2$/s white light after the end of the night (EoN) under ambient-air $CO_2$ concentrations. Different
letters indicate significant statistical differences among genotypes for the given time point for $P < 0.05$ determined by one-way ANOVA with post hoc Tukey's test (E–H).
Metabolites are from entire rosettes of WT, *stp1-1*, *stp4-1*, and *stp1stp4* plants at the end of the night (EoN) and end of the day (EoD) in a 12-h light/12-h dark cycle. FW,
fresh weight. Asterisk (*) indicates significant statistical difference between time points for the given genotype. Different letters indicate significant statistical differences
among genotypes for the given time point for $P < 0.05$ determined by one-way ANOVA with post hoc Tukey's test.

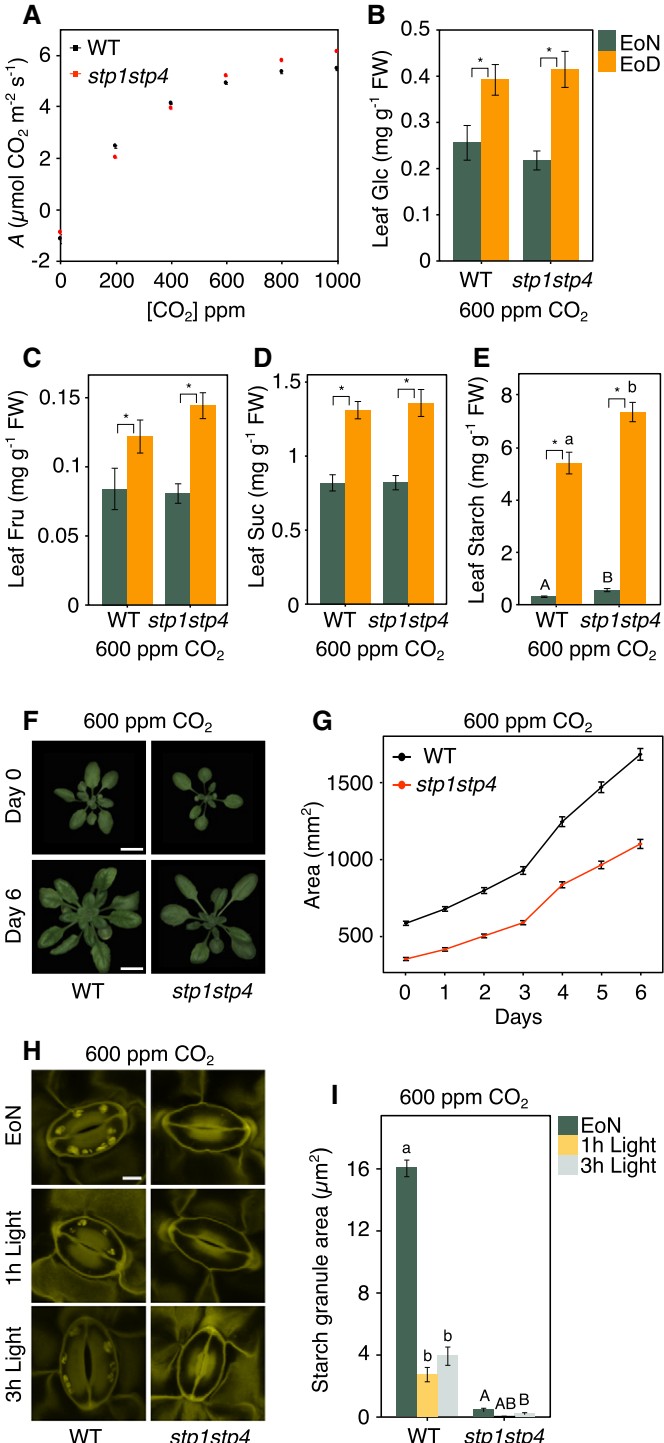

**Figure 4.  Elevated levels of CO₂ rescue the *stp1stp4* photosynthetic growth phenotype.**

A   Photosynthetic CO₂ assimilation (*A*) in dark-adapted WT and *stp1stp4* plants in response to a step increase in CO₂ concentrations from 0 to 1,000 ppm. Data shown are means ± SEM; *n* = 3 per genotype.
B   Quantification of leaf glucose (Glc). Data for two independent experiments are shown; means ± SEM; *n* ≥ 17 per genotype and time point.
C   Quantification of leaf fructose (Fru). Data for two independent experiments are shown; means ± SEM; *n* ≥ 17 per genotype and time point.
D   Quantification of leaf sucrose (Suc). Data for two independent experiments are shown; means ± SEM; *n* ≥ 17 per genotype and time point.
E   Quantification of leaf starch. Data for two independent experiments are shown; means ± SEM; *n* ≥ 17 per genotype and time point.
F   Representative Red Green Blue (RGB) images of 3-(day 0) and 4-week-old (day 6) WT and *stp1stp4* plants grown under 600 ppm CO₂. Scale bar, 10 μm.
G   Projected rosette area over the phenotyping period. Data shown are from two independent experiments; means ± SEM; *n* ≥ 24 per genotype and time point.
H   Representative confocal laser microscopy images of propidium iodide-stained guard cell starch granules of intact leaves of WT and *stp1stp4* plants grown under 600 ppm CO₂. Scale bar, 10 μm.
I   Starch dynamics in guard cells of intact leaves of WT and *stp1stp4* plants grown under 600 ppm CO₂ at the end of the night (EoN) and after 1 and 3 h of illumination with 150 μmol/m²/s of white light. Data for two independent experiments are shown; means ± SEM; *n* = 80 individual guard cells per genotype and time point.

Data information: (B–E) Metabolites are from entire rosettes of WT and *stp1stp4* plants at the end of the night (EoN) and end of the day (EoD) in a 12-h light/12-h dark cycle. Plants were grown under 600 ppm CO₂. FW, fresh weight. Asterisk (*) indicates significant statistical difference between time points for the given genotype. Different letters indicate significant statistical differences among genotypes for the given time point for *P* < 0.05 determined by one-way ANOVA with post hoc Tukey's test. (G) Asterisk (*) indicates significant statistical difference between genotypes for the given time point for *P* < 0.05 determined by one-way ANOVA with post hoc Tukey's test. (I) Different letters indicate significant statistical differences among time points for the given genotype for *P* < 0.05 determined by one-way ANOVA with post hoc Tukey's test.

cooperate in the import of mesophyll-derived Glc at dawn (Fig 2A), delivering to guard cells the carbon sources needed to promote light-induced stomatal opening (Fig 1), guard cell starch accumulation (Fig 2B and C), and plant growth (Fig 3).

STP1 was the first higher plant hexose transporter to be characterized (Sauer *et al*, 1990; Boorers *et al*, 1994). STP1 was initially found in germinating seeds and seedling roots, where it was shown to mediate the uptake of extracellular hexoses (Sherson *et al*, 2000).

Stadler *et al* (2003) reported high expression of *STP1* also in guard cells, but could not detect phenotypic changes in *stp1* mutants, perhaps due to the low resolution of their gas exchange measurements. Heterologous expression studies in yeast cells demonstrated that STP4 is also an energy-dependent monosaccharide-H⁺ symporter (Truernit *et al*, 1996). STP4 was reported to be expressed in root tips and pollen grains (Truernit *et al*, 1996), and was recently implicated in the uptake of Glc into pollen tube, where Glc serves as a signaling molecule for pollen tube growth (Rottmann *et al*, 2018a).

Our work significantly advances knowledge on plant hexose transporters by revealing a novel function for STP1 and STP4 in Glc uptake to guard cells. Reciprocal WT/*stp1stp4* grafting experiments (Figs 1D and 3D) ruled out that the *stp1stp4* stomatal phenotype was linked to the function of these transporters in the roots, further supporting a novel role for STPs in guard cells. Reduced Glc import at dawn impaired stomatal opening (Fig 1) and caused a reduction in intracellular CO₂ concentrations (Fig EV4A), which in turn impacted on photosynthetic capacity and plant growth (Fig 3). However, these CO₂ diffusional limitations could be compensated by growing the plants in elevated ambient-air CO₂ concentrations (Fig 4).

Single *stp1* and *stp4* mutants displayed mild phenotypes. In particular, *stp1* mutants showed slightly slower stomatal opening

kinetics compared to WT, whereas *stp4* had greater overall $g_s$ amplitude (Figs 1C and EV2A, C and D). Given the opposite phenotypes of the single *stp1* and *stp4* mutants, we suggest that STP1 and STP4 are not redundant in guard cells, but rather have an intricate functional relationship. Like the distantly related Glc transporters from human (Hamill *et al*, 1999) or the SUT transporters in plants (Reinders *et al*, 2002), STP1 and STP4 may form a complex at the guard cell plasma membrane. Hetero-oligomerization of these transporters would be a fast mean to regulate sugar transport properties to guard cells and adjust the loading capacity according to supply and demand. Further studies are required to assess the potential of STP transporters to form hetero-oligomers and whether/how hetero-oligomerization affects STP transport properties.

Our discovery of a novel function for STP1 and STP4 in the regulation of stomatal opening has notable implications. Firstly, we shed new light on the unresolved issue concerning the potential sources of sugar in guard cells. In our current model, mesophyll-derived Suc is postulated to be the most important carbon source, which is broken down during stomatal opening to fuel the tricarboxylic acid (TCA) cycle (Daloso *et al*, 2015, 2016b; Medeiros *et al*, 2018). Our finding of reduced levels of Glc in *stp1stp4* guard cells at the beginning of the day (Fig 2A) provides compelling evidence that imported Glc significantly contributes to the pool of sugars during dark-to-light transitions. This observation is in line with an early study which identified a monosaccharide-$H^+$ symporter activity in pea (*Pisum sativum*) guard cell protoplasts in addition to Suc uptake (Ritte *et al*, 1999). Interestingly, in the same study, it was reported that Suc uptake was much slower than that of Glc, suggesting that guard cells import carbohydrates mainly in the form of hexoses. However, during time of high $CO_2$ assimilation and transpiration (e.g. around noon), Suc uptake rates increased significantly (Lu *et al*, 1997; Ritte *et al*, 1999). These differences imply that there is a considerable flexibility in the extent and the means by which guard cells metabolize carbohydrates. It seems likely that in the early morning guard cells predominantly import Glc, while Suc is expected to become prominent among the sugars taken up when it is released in high amounts by the mesophyll, swept to the epidermis with the transpiration stream, and accumulates in high concentrations around the guard cells.

Secondly, our work provides new exciting evidence on the role of soluble sugars in guard cell function. The defective stomatal opening of *stp1stp4*, despite the presence of high levels of Suc (Fig 2A), suggests that Glc import via STPs at dawn significantly contributes to the carbon sources fuelling guard cell metabolism for stomatal opening. Apart from playing an osmotic role, Glc can be condensed in the cytosol to make hexose phosphates. These can then enter glycolysis and produce pyruvate to feed the TCA cycle; the reduced levels of Glc and Fru found in guard cells of *stp1stp4* plants might therefore cause a decreased energy status (i.e. low ATP/ADP ratio) contributing to the impaired stomatal opening in this mutant. Hexose phosphates can also be used for the synthesis of uridine diphosphate glucose (UDP-Glc), which is a precursor of cellulose and most of the other cell wall polysaccharides, or they can be translocated to the chloroplast to make starch. The lack of starch in *stp1stp4* guard cells (as we show in Fig 2B and C), and the fact that cellulose microfibrils in the cell wall are reorganized during guard cell movement in a xyloglucan and cellulose synthesis-dependent manner (Rui & Anderson, 2016), further supports the essential

role of Glc as carbon source for guard cell metabolism regulating stomatal movements. The importance of Glc presented here is in line with latest research demonstrating that Glc is the major guard cell starch-derived metabolite which helps to accelerate stomatal opening above a baseline rate (Flütsch *et al*, 2020).

The increased quantities of Suc in *stp1stp4* guard cells could be explained by upregulation of SUC transporters in the absence of functional STP1/STP4 or by reduction in Suc consumption due to feedback inhibition. Either way, Suc might enter a futile cycle of metabolic reactions, in which Suc is re-synthesized from UDP-Glc by the activity of the sucrose synthase Susy (Robaina-Estévez *et al*, 2017). Despite synthesis/degradation/import of Suc in guard cells is a dynamic process that has been shown to control stomatal movement, the existence of such Suc futile cycle might be interpreted as the closest steady-state solution to an underlying dynamic process, in which synthesis/degradation alternate according to the need of the cell (Robaina-Estévez *et al*, 2017). Furthermore, the presence of high Suc but very low starch in *stp1stp4* mutants suggests that the pathway from Suc towards starch is nearly blocked in guard cells. The reason for such blocking is unclear. It can either result from guard cell-specific properties of the corresponding enzymes or it can be due to sequestration of Suc in the vacuole, presumably via the action of the vacuolar TST-type carriers (e.g. TMT1 and TMT2) that have been demonstrated to be capable of efficient proton-coupled Suc transport (Schulz *et al*, 2011; Jung *et al*, 2015).

In conclusion, our study demonstrates that guard cells have insufficient carbon reduction capacity and their function strongly depends on carbon supply from the mesophyll mediated by STP proteins. The tight correlation between mesophyll and guard cell carbohydrate metabolism as we now show puts Glc forward as an important metabolite connecting $g_s$ and $A$, and explains how small symplastically isolated guard cells could have critical roles in regulating $CO_2$ uptake for fine-tuned photosynthetic growth.

# Materials and Methods

### Plant material and growth conditions

*Arabidopsis thaliana* accession Columbia (Col-0) was used as wild type (WT) in this study. The transfer DNA (T-DNA) insertion lines SALK_048848 (*stp1-1*) (Yamada *et al*, 2016), SALK_139194 (*stp1-2*), SALK_049432 (*stp4-1*), SALK_091229 (*stp4-2*) (Truernit *et al*, 1996), SALK_045494 (*stp13*) (Norholm *et al*, 2006; Schofield *et al*, 2009; Yamada *et al*, 2016), SALK_41553 (*suc1*), and SALK_037223 (*suc3*) were obtained from the Nottingham *Arabidopsis* Stock Centre (NASC) (Alonso *et al*, 2003), and the homozygous lines were isolated by molecular genotyping (for primer sequences see Appendix Table S2). *stp1stp4, stp1stp13,* and *stp4stp13* double mutant lines were generated by crossing through standard techniques and isolated by molecular genotyping (for primer sequences see Appendix Table S2). Seeds were sown on soil (Profi substrat, Einheits Erde, Classic), stratified at 4°C for 3 days in the dark, and transferred to the growth chamber for 7 days. Seedlings of similar size were then transplanted into single pots and cultivated in climate-controlled chambers (KKD Hiross, CLITEC Boulaguiem, Root, Switzerland; Fitoclima 1200 or Fitoclima 2500, Aralab, Rio de Mouro, Portugal) under 12 h/12 h light/dark photoperiod, with a

temperature of 21°C/19°C, a relative humidity of 45%/55%, and an irradiance of 150 μmol/m$^2$/s using a combination of white (Osram Biolux) and purple (Osram Fluora) halogen lamps (in KKD Hiross) or LED tubes (in Fitoclima 1200) or panels (in Fitoclima 2500). In the case of the high $CO_2$ experiments, plants were cultivated in the Fitoclima 2500 climate chamber equipped with an external $CO_2$ gas tank. Unless otherwise stated, experiments were performed with 4-week-old non-flowering plants.

## High-throughput phenotypic characterization

The high-throughput phenotypic characterization of WT and *stp* plants was carried out at Photon Systems Instruments (PSI) Research Center (Drásov, Czech Republic).

### *Plant growth conditions*

Plants were cultivated as described in (Awlia *et al*, 2016) with the following modifications. Seeds were sown on soil (Substrate 2, Klasmann-Deilmann GmbH, Germany), stratified at 4°C for 3 days in the dark, and transferred to a climate-controlled chamber (FytoScope FS_WI, PSI, Drásov, Czech Republic) under 12 h/12 h, 22°C/20°C light/dark photoperiod with a relative humidity of 60% and an irradiance of 150 μmol/m$^2$/s (cool-white LED and far-red LED). Seven days after stratification (DAS), seedlings of similar size were transplanted into single pots prepared the day before with 60 g of sieved soil and watered up to the maximum soil water holding capacity. Plants were cultivated in the climate-controlled chamber as described above. 18 and 20 DAS plants were watered up to 60% soil water content using the automated weighing and watering unit of PlantScreen™ Compact System (PSI, Drásov, Czech Republic).

### *High-throughput automated non-invasive phenotyping platform in controlled environment*

Plant phenotypic measurements were performed using Plant-Screen™ Compact System installed in controlled environment (FytoScope FS_WI, PSI, Drásov, Czech Republic). The platform is equipped with four robotic-assisted imaging units, an acclimation tunnel, and a weighing and watering unit. Plants set in trays are transported to the individual units by conveyor belts. The system is located in a climate-controlled chamber with a temperature of 23°C $\pm$ 1°C and a relative humidity of 55% $\pm$ 5%.

### *Phenotyping protocol and imaging sensors*

A total of 10 plants per genotype were randomly distributed in specific trays of 20 pots each. Plant imaging started 21 DAS (day 0, 3-week-old plants) and continued until 28 DAS (day 7, 4-week-old plants). Plants were imaged daily using the following protocol. Briefly, plants were manually transferred from the climate-controlled growth chamber to the conveyor belt of the acclimation tunnel of PlantScreen™ Compact System and adapted for 30 min to the controlled environment under an irradiance of 150 μmol/m$^2$/s (cool-white LED and far-red LED). Subsequently, plants were automatically phenotyped for around 2 h using thermal imaging, Red Green Blue (RGB) imaging, kinetic chlorophyll fluorescence imaging, and hyperspectral imaging in the listed order. At the end of the phenotyping protocol, plants were manually moved back to the climate-controlled growth chamber until the subsequent day. Using the automatic timing function of PlantScreen™ Scheduler (PSI,

Drásov, Czech Republic), the phenotyping protocol was programmed to start always at the same time of the diurnal cycle (after 2 h of illumination in the climate-controlled growth chamber). The acquired images were automatically processed using Plant Data Analyzer (PSI, Drásov, Czech Republic), and the raw data exported into CSV files were provided as input for analysis.

RGB imaging and processing were carried out as described in (Awlia *et al*, 2016) with the following modifications. Plant growth rate was calculated by fitting an exponential function to the interval of the projected leaf area increase over time (day 0 – day 7). For leaf greenness evaluation, 9 hues of green were automatically generated using as input RGB images captured during the phenotyping period (day 0 – day 7). After a preliminary leaf greenness analysis, the 5 most representative hues were selected and used to estimate the variations in rosette colors.

Kinetic chlorophyll fluorescence imaging described in (Awlia *et al*, 2016) was optimized using a single photon irradiance level of 440 μmol/m$^2$/s with a duration of 240 sec in the light curve protocol to quantify the rate of photosynthesis.

A thermal infrared camera (FLIR A615, FLIR Systems Inc.) with a resolution of 640 × 480 pixels mounted on a robotic arm was employed to automatically acquire top-view infrared images. The trays were automatically transported from the acclimation tunnel to the thermal imaging cabinet. Single snapshot image of one tray was acquired, and leaf surface temperature of each plant was automatically extracted with Plant Data Analyzer by pixel-by-pixel integration of values across the entire rosette. In order to subtract background from plant tissue, binary masks resulting from RGB image analysis were used. To minimize the influence of the environmental variability and the difference in the image acquisition timing among individual trays, the raw temperature of each plant (°C) was normalized by the average temperature (°C) of all plants present in the corresponding tray. The values in Fig 1B are shown as normalized leaf surface temperature.

Visible-near-infrared (VNIR) hyperspectral camera HC-900 Series (PSI, Drásov, Czech Republic) was used to acquire the spectral reflectance profiles of each plant. HC-900 camera operates in line scan mode in a wavelength range of 350–900 nm with a spectral resolution of 0.8 nm FWHM. The camera is mounted on robotic arm with implemented halogen tube light source (600 W) for homogenous and spectrally appropriate sample illumination during image acquisition. Prior to each measurement, two calibration measurements were performed automatically: dark current and radiometric using reflectance standard. Acquired hyperspectral data were processed using pixel-by-pixel analysis implemented in Hyperspectral Analyzer (PSI, Drásov, Czech Republic), featuring radiometric and dark noise calibration, background subtraction, and automated vegetation indices computation.

## Leaf and guard-cell-enriched epidermal peel RNA isolation and qPCR

To extract RNA from leaf material, one full rosette per genotype, corresponding to one biological replicate, was harvested at the end of the night and immediately frozen in liquid nitrogen. To extract RNA from guard-cell-enriched epidermal peels, the middle veins of fully developed leaves from 12 rosettes per genotype, corresponding to one biological replicate, were excised at the end of the night at

4°C in the dark and the remaining leaf material was blended in 100 ml ice-cold water using a kitchen blender (Philips, Avance Collection). The blended material was filtered through a 200 μm nylon mesh, and the remaining epidermal peels were dried, collected in a tube, and immediately frozen in liquid nitrogen. Subsequently, leaf and guard-cell-enriched epidermal peel materials were ground into a fine powder with a ball mill (Mix Mill MM-301, Retsch). Two or three biological replicates per genotype were harvested for one experiment. Two independent experiments were performed for each extraction (leaves and guard cell-epidermal peels).

Total RNA was isolated from 30 mg of grinded tissue using the SV Total RNA Isolation System (Promega) according to the manufacturer's instructions. RNA quantity and quality were determined using NanoDrop ND-1000 Spectrophotometer (Thermo Scientific). 1 μg of RNA was used for first-strand synthesis of cDNA using the M-MLV Reverse Transcriptase RNase H Minus Point Mutant (Promega) and oligo(dT)$_{15}$ primer (Promega). Transcript levels were determined by qPCR using SYBR Green PCR Master Mix (Applied Biosystems) with the 7500 Fast Real-Time PCR System (Applied Biosystems). qPCR reactions were performed in triplicates. *ACT2* was used as housekeeping gene for normalization. Transcript levels were calculated using the comparative $C_T$ ($\Delta\Delta C_T$) method (Livak & Schmittgen, 2001). Raw data ($C_T$ values) processing and error calculation were performed according to Applied Biosystems guidelines (http://assets.thermofisher.com/TFS-Assets/LSG/manuals/cms_042380.pdf). Primer sequences and PCR amplification efficiencies are listed in Appendix Table S2.

### Leaf chlorophyll quantification

To quantify chlorophyll from leaf material, one full rosette per genotype, corresponding to one biological replicate, was harvested at the end of the night and immediately frozen in liquid nitrogen. Entire rosettes were ground in liquid nitrogen using mortar and pestle. Chlorophyll from approximately 100 mg of ground plant material was extracted at −20°C for 2 h in pre-cooled 90% acetone/10% 0.2 M Tris–HCl pH 8. The supernatant was diluted 1:20 in chlorophyll extraction buffer. Samples were transferred to a glass cuvette, and the OD at 649 and 665 nm was measured spectrophotometrically. Chlorophyll (Chl) *a*, *b* and *a+b* concentrations (μg/ml) were calculated according to the following formulas:

$$Chl\ a = 11.63 * OD_{665} - 2.39 * OD_{649}$$

$$Chl\ b = 20.11 * OD_{649} - 5.18 * OD_{665}$$

$$Chl\ a + b = 6.45 * OD_{665} + 17.72 * OD_{649}$$

Three biological replicates per genotype were harvested for one experiment. Three independent experiments were performed.

### Gas exchange measurements

For gas exchange measurements, plants were grown in a Fitoclima 2500 under 8 h/16 h light/dark photoperiod, with a temperature of 21°C/19°C, a relative humidity of 45%/55%, and an irradiance of 150 μmol/m$^2$/s. Whole-plant gas exchange measurements were

carried out using the LI-6400XT System (LI-COR Biosciences) equipped with the 6400-17 whole-plant *Arabidopsis* chamber and the integrated 6400-18A RGB light source. To prevent any $CO_2$ diffusion and water vapor from the soil, pots containing *Arabidopsis* were sealed with clear film. Stomatal conductance ($g_s$) and net $CO_2$ assimilation (*A*) were measured at 22°C ± 2°C, with a relative humidity of 50% and a $CO_2$ concentration of 400 ppm. The plant was equilibrated in darkness within the chamber until all the parameters had stabilized (30 min). After the reading was constant for 10 min, an irradiance of 150 μmol/m$^2$/s was applied to the rosette for 2 h, followed by 30 min of darkness. The parameters were recorded every min. $g_s$ and *A* were normalized by subtracting the values at the end of the night (set as 0 = $g_{initial}$ or $A_{initial}$). At least three independent plants per genotype were measured on different days starting at the same time of the diurnal cycle (end of the night). The time point after 2 h of illumination corresponds to the same time of the diurnal cycle at which thermal images were taken. Rosette area was determined using ImageJ version 1.48 (NIH USA, http://rsbweb.nih.gov/ij/). To evaluate the effect of increasing concentrations of ambient-air $CO_2$ on *A*, photosynthetic $CO_2$ assimilation under irradiance of 150 μmol/m$^2$/s was measured by applying a step increase in $CO_2$ concentrations (from 0 to 1,000 ppm) after plants reached steady-state conditions for *A*.

### Guard cell starch quantification

Epidermal peels were manually prepared at the end of the night and after 1 and 3 h of illumination. Guard cell starch granules were stained and fixed as described in (Flütsch *et al*, 2018). Guard cell starch granules were visualized using confocal laser-scanning microscope Leica TCS SP5 (Leica Microsystems), and their area was quantified using ImageJ version 1.48 (NIH USA, http://rsbweb.nih.gov/ij/). Four biological replicates per genotype were measured for each time point for one experiment. Three independent experiments were performed.

### Leaf starch and soluble sugar quantification

Starch and soluble sugars (glucose, fructose, and sucrose) were extracted as described in (Thalmann *et al*, 2016) from entire rosettes harvested at the end of the night or at the end of the day. Leaf starch content was quantified as described in (Thalmann *et al*, 2016). Leaf soluble sugars were quantified based on the protocol for quantification of root soluble sugars described in (Thalmann *et al*, 2016) using as starting material 15 μl of neutralized soluble fraction obtained from the initial perchloric acid extraction. At least eight biological replicates per genotype were measured for each time point.

### Guard cell soluble sugar quantification

To extract soluble sugars from guard-cell-enriched epidermal peels, six rosettes per genotype, corresponding to one biological replicate, were collected at the end of the night or after the plants were exposed to white light of 150 μmol/m$^2$/s for 40 min and the petiole was removed using scissors. The remaining leaf material was blended in 100 ml ice-cold water using a kitchen blender (Philips, Avance Collection). The blended material was filtered through a

200 μm nylon mesh, and the remaining epidermal peels were dried, collected in a tube, and immediately frozen in liquid nitrogen. To remove residual sugars from the guard cell apoplast, the samples were washed with 2L of MilliQ water (Daloso *et al*, 2015) and refrozen in liquid nitrogen. Afterward, guard-cell-enriched epidermal peel materials were ground into a fine powder with a ball mill (Mix Mill MM-301, Retsch). Six biological replicates per genotype and time point were harvested for one experiment. Two independent experiments were performed.

Soluble sugars were extracted as described in (Thalmann *et al*, 2016). After the extraction, the samples were lyophilized in a freeze-dryer (Lyovac GT1, Lybold) and resuspended in 60 μl of MilliQ water.

Leaf soluble sugars were quantified based on the protocol for quantification of root soluble sugars described in (Thalmann *et al*, 2016) using 50 μl of neutralized soluble fraction obtained from the initial perchloric acid extraction as starting material.

### *Arabidopsis* grafts

Sterilized seeds were germinated on half-strength MS vertical plates, containing 2% (w/v) phytoagar and sealed with 3M™ Micropore™ Tape (3M Company). After 2 days of stratification at 4°C in the dark, plates were transferred to a climate-controlled chamber (Konstantraum 5, University of Zurich, Switzerland) under 16 h/8 h, light/dark photoperiod with a constant temperature of 21°C and an irradiance of 70 μmol/m$^2$/s (Osram Fluora L58W/77 and Osram Biolux L58W/965 in a 1:1 ratio). Grafts of WT and *stp1stp4* plants were generated by micrografting technique (Turnbull *et al*, 2002) under a stereomicroscope (Nikon, SMZ1500) using 3/4-day-old seedlings. Micro-scissors with angled tip and a cutting edge of 2.5 mm (Fine Science Tools) were used to perform a neat horizontal cut in the upper region of the hypocotyl of the donor lines WT and *stp1stp4* plants. A graft union between shoot and root of the donor lines was performed to generate the self-grafted donor lines WT/WT and *stp1stp4*/*stp1stp4*, and the reciprocal grafted lines WT/*stp1stp4* and *stp1stp4*/WT. Grafted plants were grown horizontally for 3–4 days in a climate-controlled chamber (Fytoscope FS130, Photon Systems Instruments, PSI, Drásov, Czech Republic) located next to the stereomicroscope to minimize movements and promote the rapid formation of the graft union. Grafted plants were grown for additional 3 days vertically under the same conditions as described above. Grafts were monitored daily. 6–7 days after grafting, successful grafts (robust graft connection and no adventitious roots) were transplanted in soil and grown in a Fitoclima 2500 or KKD Hiross under 8 h/16 h, 21°C/19°C, light/dark photoperiod with a relative humidity of 45%/55% and an irradiance of 150 μmol/m$^2$/s. Approximately 5–6 weeks after grafting, grafted plants were used to perform gas exchange measurements as described above. Subsequently, roots of grafted plants were excised and collected for molecular genotyping in order to validate the success of the grafting procedure.

### Data and statistical analysis

Data were processed using specifically developed *R* scripts (R Core Team, 2015). Statistical differences between genotypes and time points were determined by one-way analysis of variance (ANOVA) with *post hoc* Tukey's Honest Significant Difference (HSD) test ($P < 0.05$) performed using appropriate *R* scripts (R Core Team, 2015). Data are displayed as means ± SEM.

## Data availability

Material and data set produced and analyzed for this study are available upon request to the corresponding author. Sequence data from this study can be found in the *Arabidopsis* Genome Initiative or GenBank/EMBL databases under the following accession numbers: ACT2, AT3G18780; BAM3, AT4G17090; KAT1, AT5G46240; MYB60, AT1G08810; STP1, AT1G11260; STP4, AT3G19930; STP13, AT5G26340; SUC1, AT1G71880 and SUC3, AT2G02860.

**Expanded View** for this article is available online.

## Acknowledgements

We thank Diana Pazmino for technical support, Patricie Pakostová and Jaromír Pytela for plant material preparation, Zbyněk Pospíchal and Plant-Screen™ systems development team for technical support and optimization of phenotyping platform, Petr Polach for re-processing raw images, Magdalena M. Julkowska for providing R scripts, Tracy Lawson and Enrico Martinoia for helpful discussion. We further thank Michele Moles and Luca Distefano for help with preparation of guard cell-enriched material, Cyril Zipfel and Stefan Hörtensteiner from University of Zürich for providing us with growth chambers and laboratory equipment during our transition to ETH Zürich. Data produced in this paper were partially generated in collaboration with the Genetic Diversity Centre (GDC), ETH Zurich. This work was supported by European Union's Seventh Framework Program for research, technological development and demonstration under grant agreement no PITN-GA-2013-608422—IDP BRIDGES (to D.S.), the Swiss National Science Foundation SNSF-Grant 31003A_166539 and SNSF-Grant 310030_185241 (to D.S.), European Union's Horizon 2020 research and innovation program under grant agreement no 722338—PlantHUB (to D.S.), the ETH Zürich and the University of Zürich.

## Author contributions

DS conceptualized research; DS, KP, SF, and AN designed experiments; SF, AN, FC and MTh performed research; SF, AN, JF, MTr and KP analyzed the data; DS, SF and AN wrote the paper.

## Conflict of interest

The authors declare that they have no conflict of interest.

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
