## [Review Process File · EMBO Reports]

Glucose uptake to guard cells via STP transporters provides carbon sources for stomatal opening and plant growth

Diana Santelia, Sabrina Flütsch, Arianna Nigro, Franco Conci, Jiří Fajkus, Matthias Thalmann, Martin Trtílek, and Klára Panzarová

DOI: [10.15252/embr.201949719](https://doi.org/10.15252/embr.201949719)

Corresponding author(s): *Diana Santelia (dsantelia@ethz.ch)*

Review Timeline:

Submission Date:	20th Nov 19
Editorial Decision:	22nd Dec 19
Revision Received:	24th Mar 20
Editorial Decision:	30th Apr 20
Revision Received:	8th May 20
Accepted:	13th May 20

Editor: *Deniz Senyilmaz Tiebe*

Transaction Report:

Dear Dr. Santelia,

Thank you for the submission of your research manuscript to our journal, which was now seen by three referees, whose reports are copied below.

As you can see, the referees express interest in the analysis. However, they also raise a number of concerns that need to be addressed to consider publication here. I find the reports informed and constructive, and believe that addressing the concerns raised will significantly strengthen the manuscript.

Given these constructive comments, we would like to invite you to revise your manuscript with the understanding that the referee concerns (as in their reports) must be fully addressed and their suggestions taken on board. Please address all referee concerns in a complete point-by-point response. Acceptance of the manuscript will depend on a positive outcome of a second round of review. It is EMBO reports policy to allow a single round of revision only and acceptance or rejection of the manuscript will therefore depend on the completeness of your responses included in the next, final version of the manuscript.

1. A data availability section providing access to data deposited in public databases is missing (where applicable).
2. Your manuscript contains statistics and error bars based on $n=2$ or on technical replicates. Please use scatter plots in these cases.

Supplementary/additional data: The Expanded View format, which will be displayed in the main HTML of the paper in a collapsible format, has replaced the Supplementary information. You can submit up to 5 images as Expanded View. Please follow the nomenclature Figure EV1, Figure EV2 etc. The figure legend for these should be included in the main manuscript document file in a section called Expanded View Figure Legends after the main Figure Legends section. Additional Supplementary material should be supplied as a single pdf labeled Appendix. The Appendix includes a table of content on the first page with page numbers, all figures and their legends. Please follow the nomenclature Appendix Figure Sx throughout the text and also label the figures according to this nomenclature. For more details please refer to our guide to authors.

2) individual production quality figure files as .eps, .tif, .jpg (one file per figure).

3) a .docx formatted letter INCLUDING the reviewers' reports and your detailed point-by-point responses to their comments. As part of the EMBO Press transparent editorial process, the point-by-point response is part of the Review Process File (RPF), which will be published alongside your paper. For more details on our Transparent Editorial Process, please visit our website:

<https://www.embopress.org/page/journal/14693178/authorguide#transparentprocess>

4) a complete author checklist, which you can download from our author guidelines (). Please insert information in the checklist that is also reflected in the manuscript. The completed author checklist will also be part of the RPF.

5) Please note that all corresponding authors are required to supply an ORCID ID for their name upon submission of a revised manuscript (). Please find instructions on how to link your ORCID ID to your account in our manuscript tracking system in our Author guidelines ().

6) We replaced Supplementary Information with Expanded View (EV) Figures and Tables that are collapsible/expandable online. A maximum of 5 EV Figures can be typeset. EV Figures should be cited as 'Figure EV1, Figure EV2' etc... in the text and their respective legends should be included in the main text after the legends of regular figures.

7) We would also encourage you to include the source data for figure panels that show essential data.

Numerical data should be provided as individual .xls or .csv files (including a tab describing the data). For blots or microscopy, uncropped images should be submitted (using a zip archive if multiple images need to be supplied for one panel). Additional information on source data and instruction on how to label the files are available .

8) Regarding data quantification, please ensure to specify the name of the statistical test used to generate error bars and P values, the number (n) of independent experiments underlying each data point (not replicate measures of one sample), and the test used to calculate p-values in each figure legend. Discussion of statistical methodology can be reported in the materials and methods section, but figure legends should contain a basic description of n, P and the test applied.

Please note that error bars and statistical comparisons may only be applied to data obtained from

at least three independent biological replicates.
Please also include scale bars in all microscopy images.

I look forward to seeing a revised version of your manuscript when it is ready. Please let me know if you have questions or comments regarding the revision.

Kind regards,

Deniz Senyilmaz Tiebe

Deniz Senyilmaz Tiebe, PhD
Editor
EMBO Reports

Referee #1:

The manuscript "Glucose uptake to guard cells via STP transporters provides carbon sources for stomatal opening and plant growth" by Flütsch et al. highlight the identification and characterization of the monosaccharide/proton symporters Sugar Transport Protein 1 and 4 (STP1 and STP4) as major hexose sugar transporters in guard cells of *Arabidopsis thaliana*. The authors provide strong evidence that STP1 and STP4 are required for the import of glucose into guard cells to promote light-induced stomatal opening, starch accumulation in guard cells and plant growth by employing a vast majority of genetic, biochemical and physiological approaches. Flütsch et al. show that the stomatal phenotype of *stp1 stp4* lies in guard cell expression and function and is independent from their function in roots by using elegant grafting experiments.

Major critical points:

Stomatal conductance and assimilation rate time courses figures show differential subtractive responses (Figs. 1C, 1D, 3C, 3D, 4A, EV3). Graphs showing the data without subtraction should be included in the manuscript figures, so that any effects of differences in steady state values are easily visible.

Minor critical points:

1) Transcript level reduction of *stp4* (Fig. EV2) lack strong evidence due to high error bars

- 2) Fig 1D: I recommend repeating this experiment to strengthen the data since n is only {greater than or equal to}2.
- 3) A second allele of the *stp1 stp4* double mutant or complementation would be helpful and support the present data.
- 4) Analysis of soluble sugars in Fig. 2A lacks data for single *stp1* and *stp4* mutants, especially since starch accumulation in guard cells was quantified for both single mutants in 2C.
- 5) Page 7, line 180-182: "In WT guard cells, the levels of Glc [...] decreased in response to the light treatment". This conclusion is not supported by the data shown in Fig. 2A (not a significant difference).
- 6) Why does Figure 1B show normalized data only?
- 7) When reporting C_i data it should be made clear that C_i is calculated and not directly measured.

Referee #2:

Comments on the manuscript "Glucose uptake to guard cells via STP transporters provides carbon sources for stomatal opening and plant growth by Flütsch et al.

Stomata at the leaf surface are built by a pair of specific epidermal cells, named guard cells. Due to changes in the osmotic potential of guard cells stomata either open or close, and this process is critical for control of water loss and CO₂ uptake, required for photosynthesis.

Although guard cells are in the focus of scientific work for decades, our knowledge of their primary metabolism is far from being sufficient. This, however, is important to analyze, since proper guard cells function not only influence plant characteristics in general, but is also of importance for yield of crop plants, especially under challenging environmental conditions.

In this manuscript (MS), the authors have identified that the plasma membrane located sugar transporter proteins (STP) 1 and 4 are both involved in fueling guard cells with sugars required to maintain metabolic integrity. In the course of this work it was shown that both genes are expressed in guard cells, that corresponding single knock-out plants do not exhibit modified stomata properties, while a double mutant (*stp1stp4*) displays impaired opening of stomata (Fig. 1), leading to a decreased overall growth rate. In *stp1stp4* double knock-out plants the guard cells contain strongly decreased monosaccharide (Glc + Fru) and starch levels, while sucrose concentrations are increased (Fig. 2). Interestingly, impaired growth of *stp1stp4* plants is gradually absent at increasing external CO₂ concentrations (Fig. 4) which confirms that limited entry of carbon dioxide into the leaf of double mutants is causative for this growth defect. Reciprocal grafting experiments finally confirmed that altered guard cell properties are specific and do not represent a pleiotropic effect caused by unintended modifications in the roots (which easily could affect leaf properties).

This paper fully convinces me in terms of its scientific originality, the quality of the experiments conducted, the scientific documentation coupled to a concise, but fully comprehensive discussion.

Accordingly, I have only minor comments which should be considered by the authors:

1. According to the sugar data provided in Fig. 2A, *stp1stp4* mutants contain low monosaccharides but high sucrose in guard cells ("guard cell-enriched epidermal cells"). In addition, mutant stomata hardly contain starch in guard cell chloroplasts (Fig. 2B). The authors speculate on the interaction

between sucrose and monosaccharides (Glc+Fru) in guard cells (pages 13-16) but did not mention that this observation (high suc but low starch) indicates that the pathway from sucrose towards starch seems to be (nearly) blocked in guard cells. The reason for such blocking is unclear and can be due to specific properties of corresponding enzymes or due to sequestration of sucrose in the vacuole. Latter organelles are known to be able to import sucrose via TST-type carriers (see e.g. Schulz et al. 2011, *Plant J.*, 68: 129-136 or Jung et al. 2015, *Nat. Plants* 1:14001). I think it is worth to mention such hypothesis.

2. The data presented clearly document that decreased activity of both, STP1 and STP4 leads to impaired stomatal opening. Given that sucrose to starch conversion seems to be "blocked" it appears at least possible that low Glc+Fru levels in double mutant guard cells cause a decreased energy status (low ATP/ADP ratio). When having in mind that stomatal opening is an energy consuming process it might be that such low cellular energy status contributes to impaired stomatal opening. The authors might integrate this possibility into the discussion.

3. Lines 131-133. Three sentences start with "STPs". Please, rephrase.

The MS reports novel findings on a physiological key process in plants, namely regulation of stomatal aperture. This process is important from the point of basic research but also exerts ecological- and economic impacts.

I have no doubt that these findings will be of interest for a broad readership.

Referee #3:

The manuscript "Glucose uptake to guard cells via STP transporters provides carbon sources for stomatal opening and plant growth" by Flütsch et al, describe the role of sugar (glucose) transport to guard cells and its impact on regulation of stomatal opening, photosynthesis and ultimately plant growth.

I have a few major and minor comments.

Major comments:

1. Only single alleles for tDNA mutants are used throughout all experiments. As there are several examples in the literature of mutant phenotypes not being linked to the tDNA insertion; have the authors tested additional alleles or considered genetic complementation? This is especially relevant for the double mutant *stp1 stp4*.

2. I do not understand parts of the Discussion, especially lines 320-328. Here the authors claim that double mutant *stp1 stp4* have the same phenotype as the single mutants *stp1* and *stp4*. This is opposite to all data presented in the manuscript (for example Figs 1B, 1C, 2C, 3B, 3F, 3G, 3H, 3J), where the double mutant is clearly much more impaired than the single mutants. I suggest re-writing this paragraph, or to be more specific and refer to which experiment that would show the similar phenotype between single and double mutants.

3. I do not understand the results related to table EV5 and lines 232-238. I agree with authors about the results shown in the table: the double mutant *stp1 stp4* is strongly impaired in growth and that the single mutants *stp1*, *stp4* and *stp13* are mildly impaired. But why are the double mutants *stp1 stp13* and *stp4 stp13* similar to wildtype? My expectation would have been see to impaired growth in these double mutants. Two possible explanations for the lack of phenotype in

these double mutants: 1. Some sort of compensation mechanism when *stp13* is knocked out? 2. More alarmingly and related to major comment 1 above, if the mutant phenotype is not linked to the tDNA insertion, then in these double mutants the unlinked mutation causing the phenotype in *stp1* and *stp4*, is not present in the double mutants. Again, my question is whether the authors have tested additional tDNA alleles or performed genetic complementation.

4. Figure EV9. The legend lacks information of what is shown in the figure. For example, the (B) panel is not defined in the legend (or perhaps the A should be replaced with B?). The colored lines in the B panel are not defined (which mutants is which?).

Minor comment:

5. Figure EV4. It is good practice to include negative (water) control in PCR genotyping.

EMBOR-2019-49719V2: point-by-point response to the referees

We are pleased that our manuscript was well-received by all reviewers. We have taken the referees' suggestions into full consideration and have integrated them in our revised manuscript accordingly. We respond to their comments in detail below.

Referee #1:

The following major critical point was raised by referee #1:

1. Stomatal conductance and assimilation rate time courses figures show differential subtractive responses (Figs. 1C, 1D, 3C, 3D, 4A, EV3). Graphs showing the data without subtraction should be included in the manuscript figures, so that any effects of differences in steady state values are easily visible.
1. **Response:** We thank referee #1 for raising her/his concern about the normalized g_s and A data. Our reasoning behind normalizing g_s and A data was that we were interested to follow the absolute amplitude along with the speed of increases in g_s and A between genotypes. Normalized g_s and A data allow a more convenient comparison of these parameters, as there is an intrinsic variability in g_s responses within and between genotypes, especially in Arabidopsis.
In the current revised version of our manuscript we have included the raw data for g_s and A with errors and statistics (one-way ANOVA) in the main figures (Figs 1,3 and 4). We agree that they should be available for the readers. As we are convinced of the benefit of the normalized data for fast comparisons of absolute amplitudes and opening kinetics between genotypes, we have kept the normalized data in the Expanded View Figures (Figs EV2 and EV4).

The following seven minor critical points were raised by referee #1:

1. Transcript level reduction of *stp4* (Fig. EV2) lack strong evidence due to high error bars
2. **Response:** As recommended by referee #1 we have performed a repetition of the gene expression analysis of *STP4* in the leaves of the *stp4-1* mutant compared to WT leaves. With the addition of this experiment we were able to reduce the error bars (fold change range) from 0.42 to 0.16. Referee #1 will find the new data in Figure EV1C.
2. Fig 1D: I recommend repeating this experiment to strengthen the data since n is only {greater than or equal to} 2.
3. **Response:** Based on the recommendation of referee #1, we set up a small-scale grafting experiment for WT/*stp1stp4* and we managed to generate

another true grafted plant. We repeated the gas exchange measurement with this plant and added the data to the Figures 1D, 3D, EV2B and EV4E.

3. A second allele of the *stp1 stp4* double mutant or complementation would be helpful and support the present data.

4. **Response:** We appreciate the suggestion of referees #1 and #3 to consider using a second allele for the *stp1stp4* double mutant or complementation lines. We agree that there have been cases where the T-DNA insertion was not linked to the observed phenotype, but rather had pleiotropic effects. Given the limited time for the revision of our manuscript (3 months), we have obtained second mutant alleles for *stp1* and *stp4*, herein named *stp1-2* and *stp4-2*. On these newly isolated alleles, we repeated the gas exchange measurements. As referees #1 and #3 will see in Figures EV2C, EV2D, EV4F and EV4G, the second allele mutants behaved very similar to the primary mutants used in this study. Specifically, like *stp1-1* mutant plants, *stp1-2* plants displayed reduced overall stomatal conductance compared to WT. Furthermore, in the normalized data we detect similarly reduced opening velocity for both mutant alleles of STP1. Finally, the photosynthetic assimilation is affected to a similar extent in both mutant alleles. Similar results were obtained for the additional allele of the *stp4* mutant. The overall stomatal conductance of *stp4-2* plants was similarly lower compared to WT as in *stp4-1* mutants. In addition, *stp1-4* plants show a characteristic fast and strong increase in stomatal conductance during the first hour of illumination compared to WT. We observed the same pattern in the *stp4-2* mutant. Consequently, the photosynthetic assimilation was mildly elevated in comparison to WT plants in both mutant alleles of STP4. Based on these data, we are confident that the observed phenotype in *stp1stp4* plants is a direct consequence of the insertion in the respective gene loci and not caused by pleiotropic effects.

To complete the dataset for the second allele mutants, we further performed gene expression analyses to confirm the downregulation of the respective gene in leaves. Figure EV1D of the revised manuscript shows that *stp1-2* mutants contain only about 40% of *STP1* transcripts and *stp4-2* mutants about 60% of *STP4* transcripts. Furthermore, the second allele of the *stp4* mutant has been used for research previously and has been published in Truernit *et al.*, 1996, *Plant Cell*.

4. Analysis of soluble sugars in Fig. 2A lacks data for single *stp1* and *stp4* mutants, especially since starch accumulation in guard cells was quantified for both single mutants in 2C.

5. **Response:** Referee #1 is correct in stating that for consistency we should include guard cell soluble sugar data for *stp1-1* and *stp4-1* single mutants. The reasoning behind performing the guard cell sugar quantification only on WT and *stp1stp4* plants was that we did not detect a strong phenotype in stomatal function in the single mutants compared to WT plants. Furthermore, the quantification of metabolites in guard cells of *Arabidopsis* is difficult and labor-

intensive. However, we agree with referee #1 that for completeness these data are essential, so we performed the requested additional measurements. Figure EV3B of the revised manuscript contains the results of the guard cell soluble sugar measurements in the single mutants *stp1-1* and *stp4-1*. In line with the reduced guard cell starch content at the end of the night in *stp1-1* guard cells compared to WT, we found that *stp1-1* guard cells contained reduced levels of Glc at both time points, likely explaining the reduced stomatal opening phenotype of these plants. However, the differences are not statistically significant. Further, *stp4-1* guard cells contained similar amounts of Glc as WT guard cells at both time points, which goes well along with the guard cell starch content measured in this mutant. In contrast, Fru and Suc contents were comparable between all genotypes at both time points. Since both mutants have mild stomatal phenotypes (e.g. guard cell starch, stomatal opening) this likely explains why we did not detect an over-accumulation of Suc in these mutants as it was the case for *stp1stp4*. These data further support our hypothesis that these transporters do not act redundantly, but rather have a functional relationship for the coordinate import of mainly Glc to guard cells.

5. Page 7, line 180-182: "In WT guard cells, the levels of Glc [...] decreased in response to the light treatment". This conclusion is not supported by the data shown in Fig. 2A (not a significant difference).
6. **Response:** In the revised version of the manuscript we exchanged the sentence "In WT guard cells, the levels of Glc [...] decreased in response to the light treatment" with "In WT guard cells, the levels of Glc and Suc were unaltered in response to the light treatment...". Thank you for the suggestion.
6. Why does Figure 1B show normalized data only?
7. **Response:** In order to reduce the influence of the environmental variability and the differences in the image acquisition timing among the individual plant trays to a minimum, the raw temperature of each plant was normalized by the average temperature of all plants present in the corresponding tray. However, the readers will find the raw temperature in °C for day 0 and day 7 for each genotype in Appendix Table S3.
7. When reporting C_i data it should be made clear that C_i is calculated and not directly measured.
8. **Response:** We agree that C_i is not measured directly and that it is based on a number of assumptions. We adjusted our wording in the manuscript including figure legends. Thank you for pointing it out.

Referee #2:

The following three minor comments were raised by referee #2:

1. According to the sugar data provided in Fig. 2A, *stp1stp4* mutants contain low monosaccharides but high sucrose in guard cells ("guard cell-enriched epidermal cells"). In addition, mutant stomata hardly contain starch in guard cell chloroplasts (Fig. 2B). The authors speculate on the interaction between sucrose and monosaccharides (Glc+Fru) in guard cells (pages 13-16) but did not mention that this observation (high suc but low starch) indicates that the pathway from sucrose towards starch seems to be (nearly) blocked in guard cells. The reason for such blocking is unclear and can be due to specific properties of corresponding enzymes or due to sequestration of sucrose in the vacuole. Latter organelles are known to be able to import sucrose via TST-type carriers (see e.g. Schulz et al. 2011, Plant J., 68: 129-136 or Jung et al. 2015, Nat. Plants 1:14001). I think it is worth to mention such hypothesis.
2. **Response:** Referee #2 raised a very valuable point. We thank her/him for the suggestion of including the hypothesis that Suc might be sequestered in the vacuole in our *stp1stp4* mutant, thus potentially explaining why it is not used to make starch. We make reference to the research of Schulz *et al.* 2011 in the discussion of our revised manuscript to take this hypothesis into account.
2. The data presented clearly document that decreased activity of both, STP1 and STP4 leads to impaired stomatal opening. Given that sucrose to starch conversion seems to be "blocked" it appears at least possible that low Glc+Fru levels in double mutant guard cells cause a decreased energy status (low ATP/ADP ratio). When having in mind that stomatal opening is an energy consuming process it might be that such low cellular energy status contributes to impaired stomatal opening. The authors might integrate this possibility into the discussion.
2. **Response:** According to the suggestion of referee #2, we expanded the discussion about the possible causes for the strongly affected stomatal opening response of *stp1stp4* plants, including the possibility that an altered energetic status of the double mutant guard cells might contribute to the observed phenotype. Thank you for this suggestion.
3. Three sentences start with "STPs". Please, rephrase.
3. **Response:** We reworded the corresponding passage of the manuscript. Thank you for drawing our attention to it.

Referee #3:

The following four major critical points were raised by referee #3:

1. Only single alleles for tDNA mutants are used throughout all experiments. As there are several examples in the literature of mutant phenotypes not being linked to the tDNA insertion; have the authors tested additional alleles or considered genetic complementation? This is especially relevant for the double mutant *stp1 stp4*.
1. **Response:** Please see our response 4 to referee #1 where we explain in detail the action we took to satisfy the request of both referee #1 and #3.
2. I do not understand parts of the Discussion, especially lines 320-328. Here the authors claim that double mutant *stp1 stp4* have the same phenotype as the single mutants *stp1* and *stp4*. This is opposite to all data presented in the manuscript (for example Figs 1B, 1C, 2C, 3B, 3F, 3G, 3H, 3J), where the double mutant is clearly much more impaired than the single mutants. I suggest re-writing this paragraph, or to be more specific and refer to which experiment that would show the similar phenotype between single and double mutants.
2. **Response:** We apologize for the confusing wording in the mentioned paragraph. Our intention was not to claim that *stp1stp4* double mutants have a similar phenotype as the *stp1* and *stp4* single mutants. We agree with referee #3 that the phenotype of the double mutant is consistently more pronounced compared to the single mutants. We have adjusted our wording accordingly in the discussion of the revised manuscript. Thank you for drawing our attention to it.
3. I do not understand the results related to table EV5 and lines 232-238. I agree with authors about the results shown in the table: the double mutant *stp1 stp4* is strongly impaired in growth and that the single mutants *stp1*, *stp4* and *stp13* are mildly impaired. But why are the double mutants *stp1 stp13* and *stp4 stp13* similar to wildtype? My expectation would have been to see impaired growth in these double mutants. Two possible explanations for the lack of phenotype in these double mutants: 1. Some sort of compensation mechanism when *stp13* is knocked out? 2. More alarmingly and related to major comment 1 above, if the mutant phenotype is not linked to the tDNA insertion, then in these double mutants the unlinked mutation causing the phenotype in *stp1* and *stp4*, is not present in the double mutants. Again, my question is whether the authors have tested additional tDNA alleles or performed genetic complementation.
3. **Response:** We thank referee #3 for raising her/his concern about the phenotype of the *stp1stp13* and *stp4stp13* double mutants. Since we do already not see a pronounced phenotype in the single mutants *stp4* and *stp13*, actually in many experiments they performed better than WT plants (e.g. guard cell starch contents at the end of the night, metabolites in the leaves, stomatal

conductance and photosynthetic assimilation), we started to look into a reason behind the observed phenotypes. The most realistic scenario includes that other STP isoforms partially take over the role of the missing STP. To investigate such a possibility, we examined the transcript levels of *STPs* in *stp* mutant backgrounds. To do so, we compared STP gene expression in guard cells of WT with that one in guard cells of *stp1-1*, *stp4-1* and *stp13*. As we show in Appendix Figure S6 of the revised manuscript, *STP13* is strongly upregulated in *stp1-1* guard cells, possibly partly compensating for the loss of *STP1* and contributing to the mild phenotypes of the *stp1-1* mutant. Furthermore, we demonstrate that *STP1* is highly overexpressed in *stp4-1* and *stp13* guard cells. Based on this dataset, it is likely that the phenotypes in *stp4-1* and *stp13* plants are partially due to a compensation mechanism through *STP1*. Hence, we are confident that the high gene expression levels of *STP1* explain the lack of a phenotype also in *stp1stp13* and *stp4stp13* double mutants.

4. Figure EV9. The legend lacks information of what is shown in the figure. For example, the (B) panel is not defined in the legend (or perhaps the A should be replaced with B?). The colored lines in the B panel are not defined (which mutants is which?).
4. **Response:** We apologize for the bad quality of Figure EV9 in the previous version of the manuscript. All figures of the revised manuscript have been re-edited and re-organized to meet the journals quality standard and the guidelines of the Expanded View Figures. Figure EV9 is now integrated into Figure EV5, including the legend for panel B. Furthermore, we corrected the wrong explanation in the figure legend. Thank you for pointing it out.
1. Figure EV4. It is good practice to include negative (water) control in PCR genotyping.
5. **Response:** We fully agree with referee #3 that the inclusion of water controls in molecular genotyping is standard. We have included water controls when we originally genotyped the SALK lines for *stp1-1* and *stp4-1*, but we do not show this in our final PCR.

Dear Diana,

Thank you for submitting the revised version of your manuscript. It has now been seen by two of the original referees.

As you can see, the referees find that the study is significantly improved during revision and recommend publication. Before I can accept the manuscript, I need you to address some minor points below:

- Please address the remaining minor concerns of referee #1.
- We noted that there are 2 MTs in the Author Contributions section. Rather than using the full surname, please use the 2nd initial of the surname as well.
- As per our guidelines, in the reference list, citations should be listed in alphabetical order and then chronologically, with the authors' surnames and initials inverted; where there are more than 10 authors on a paper, 10 will be listed, followed by 'et al.' Please see for further details: <https://www.embopress.org/page/journal/14693178/authorguide#referencesformat>
- The Appendix figure is should be named 'Appendix Figure S1.
- Papers published in EMBO Reports include a 'Synopsis' to further enhance discoverability. Synopses are displayed on the html version of the paper and are freely accessible to all readers. The synopsis includes a short standfirst summarizing the study in 1 or 2 sentences that summarize the key findings of the paper and are provided by the authors and streamlined by the handling editor. I would therefore ask you to include your synopsis blurb.
- In addition, please provide an image for the synopsis. This image should provide a rapid overview of the question addressed in the study but still needs to be kept fairly modest since the image size cannot exceed 550x400 pixels.
- Our production/data editors have asked you to clarify several points in the figure legends (see attached document). Please incorporate these changes in the attached word document and return it with track changes activated.

Thank you again for giving us to consider your manuscript for EMBO Reports, I look forward to your minor revision.

Kind regards,

Deniz

--

Deniz Senyilmaz Tiebe, PhD
Editor
EMBO Reports

Referee #1:

The authors have carefully revised the manuscript following the reviewer suggestions. The additional analyses of independent T-DNA lines show that T-DNA insertions in *stp1* and *stp4* are responsible for the observed phenotypes. Analysis of soluble sugars has been expanded. The efforts to generate more data with grafted plants strengthen the manuscript. Some final minor suggestions for consideration by the authors could be:

1) Thank you for adding the raw temperature imaging data in Appendix Table S3. Although differences between stp1stp4 compared with WT were statistically different, the actual differences are small. This could be indicated

2) Showing the absolute stomatal conductance data make clear that the steady state stomatal conductance of stp1-1, stp4-1 and stp1stp4 were reduced compared to wild-type plants (revised Fig. 1C+D). This could be discussed in the revised manuscript. It's still clear that the light-induced stomatal opening response was severely affected in stp1stp4 plants.

Referee #3:

The authors have answered all of my previous comments. In the future I suggest that the authors would consider making the additional double mutant stp1-2 stp4-2.

Dear Dr. Senyilmaz Tiebe,

Herewith I resubmit our revised manuscript entitled "Glucose uptake to guard cells via STP transporters provides carbon sources for stomatal opening and plant growth" for your consideration.

I am pleased that our previous version was well-received by all referees. I am also grateful to you and all the referees for the input on how to improve our work. As you will see in my "Response to referees", we have taken the referees' suggestions into full consideration and have integrated them in our revised manuscript accordingly. In particular:

1) As suggested by referee #1 and #3, we have performed additional experiments with newly isolated allele mutants of *stp1* and *stp4* (named *stp1-2* and *stp4-2*) to further demonstrate that the phenotype of *stp1stp4* double mutant is directly linked to the T-DNA insertion in the corresponding loci. The data are presented in the new Figures EV1D, EV2C, EV2D, EV4F and EV4G:

a. To confirm the downregulation of STP1 and STP4 in the second allele mutants (both knockdown), we examined gene expression in leaves of *stp1-2* and *stp4-2* (Fig EV1D). *stp1-2* plants had 60% reduction in STP1 transcripts and *stp4-2* had 40% reduction in STP4 transcripts compared to the levels of the respective genes in WT.

b. We have performed gas exchange measurements with *stp1-2* and *stp4-2* mutants under the same experimental conditions as for the primary mutants. The newly isolated allele mutants behaved nearly identical to the primary mutants used in the study. In particular, we observed that *stp1-2* mutants had reduced overall steady-state stomatal conductance compared to WT, similar to the *stp1-1* mutant (Fig EV2C and EV2D). Photosynthetic assimilation was also affected in similar ways in these two allelic mutants (Fig EV4F and EV4G). Similar results were obtained for the *stp4-2* mutant; the overall stomatal conductance was reduced to a similar extent as in *stp4-1* plants compared to WT. In addition, we observed that also *stp4-2* plants showed a characteristic fast and pronounced increase in stomatal conductance during the first hour of illumination in comparison with WT as we have previously seen for *stp4-1*. Consequently, the photosynthetic assimilation was elevated in both mutant alleles of STP4. We extensively discuss these new data in the revised manuscript.

With the presented data, we are confident that the phenotype observed in our *stp1stp4* mutant is directly linked to the T-DNA insertion in the respective STP1 and STP4 loci and not due to pleiotropic effects.

2) As suggested by referee #1, we have quantified soluble sugars in guard cells of *stp1* and *stp4* plants for completeness of guard cell metabolite measurements. We have included the resulting data in the new Figure EV3B. We observed that guard cells of *stp1-1* plants had reduced amounts of glucose at the end of the night and after 40 min of illumination compared to WT and *stp4-1*. These data are in line with the guard cell starch quantification showing that *stp1-1* guard cells have less starch at the end of the night than WT, and may also explain the slower opening phenotype of *stp1* mutant. Fructose and sucrose contents were comparable to WT guard cells in both mutants at both time points. These new data further support our conclusion that STP1 and STP4 do not act redundantly, but rather have an intricate functional relationship coordinating the import of glucose to guard cells.

3) As recommended by the referees, we have included the raw data of stomatal conductance and photosynthetic assimilation with errors and statistics in the main figures of the revised manuscript, we have repeated gene expression analysis of STP4 in *stp4-1* plants to strengthen the data, and we have performed an additional small-scale grafting experiment to reach a sample size of $n=3$ for all grafted plants. Furthermore, we integrated suggested hypotheses and publications in our revised discussion and corrected our wording according to the individual suggestions of the referees. The specific changes we made in the revised version are all listed in the "response to referees".

Thanks to your input and that of the referees, I believe our manuscript is now much improved.

I hope you will agree with me that with the included changes our revised manuscript may now be suitable for publication in EMBO Reports.

I look forward to hearing from you soon.

Yours sincerely,
Diana Santelia

Dear Diana,

Thank you for submitting your revised manuscript. I have now looked at everything and all is fine. Therefore I am very pleased to accept your manuscript for publication in EMBO Reports.

Congratulations on a nice study!

Kind regards,

Deniz

--

Deniz Senyilmaz Tiebe, PhD

Editor

EMBO Reports

Corresponding Author Name: Dr. Diana Santelia and Dr. Klára Panzarová

Manuscript Number: EMBOR-2019-49719V3